# Integrity of Cerebellar Fastigial Nucleus Intrinsic Neurons Is Critical for the Global Ischemic Preconditioning

**DOI:** 10.3390/brainsci7100121

**Published:** 2017-09-21

**Authors:** Eugene V. Golanov, Angelique S. Regnier-Golanov, Gavin W. Britz

**Affiliations:** 1Department of Neurosurgery, Houston Methodist Hospital, Houston, TX 77030, USA; gbritz@houstonmethodist.org; 2Department of Neurosurgery, University of Mississippi Medical Center, Jackson, MS 39216, USA; agolanov@gmail.com; 3Department of Pediatrics, Baylor College of Medicine, Houston, TX 77030, USA

**Keywords:** ischemic preconditioning, neuroprotection, cerebral ischemia, fastigial nucleus, stroke, hypoxic-ischemic brain injury, pathophysiology, physiology

## Abstract

Excitation of intrinsic neurons of cerebellar fastigial nucleus (FN) renders brain tolerant to local and global ischemia. This effect reaches a maximum 72 h after the stimulation and lasts over 10 days. Comparable neuroprotection is observed following sublethal global brain ischemia, a phenomenon known as preconditioning. We hypothesized that FN may participate in the mechanisms of ischemic preconditioning as a part of the intrinsic neuroprotective mechanism. To explore potential significance of FN neurons in brain ischemic tolerance we lesioned intrinsic FN neurons with excitotoxin ibotenic acid five days before exposure to 20 min four-vessel occlusion (4-VO) global ischemia while analyzing neuronal damage in Cornu Ammoni area 1 (CA1) hippocampal area one week later. In FN-lesioned animals, loss of CA1 cells was higher by 22% compared to control (phosphate buffered saline (PBS)-injected) animals. Moreover, lesion of FN neurons increased morbidity following global ischemia by 50%. Ablation of FN neurons also reversed salvaging effects of five-minute ischemic preconditioning on CA1 neurons and morbidity, while ablation of cerebellar dentate nucleus neurons did not change effect of ischemic preconditioning. We conclude that FN is an important part of intrinsic neuroprotective system, which participates in ischemic preconditioning and may participate in naturally occurring neuroprotection, such as “diving response”.

## 1. Introduction

Stimulation of the cerebellar fastigial (medial) nucleus (FN) is capable of increasing brain tolerance to damaging stimuli [1,2,3]. Volume of the infarction induced by permanent middle cerebral artery occlusion (MCAO) in FN-stimulated animals is ~50% smaller compared to sham or cerebellar dentate (lateral) nucleus (DN)-stimulated animals [4,5,6,7]. FN stimulation is equally potent in reducing damage induced by direct administration of excitotoxin (ibotenic acid, IBO) into striatum [8] and ameliorating neuronal damage following global ischemia [9]. The phenomenon of stimulation-induced neuroprotection was called central neurogenic neuroprotection to stress the neurogenic origin of the tolerance [10]. 

The salvaging effect of FN stimulation is independent of cerebral blood flow (CBF) or cerebral metabolism [4,11] and lasts for over 10 days, reaching maximum on the third day following the stimulation [9]. Neurogenic neuroprotection induced by stimulation of FN is mediated by intrinsic neurons of the nucleus. Excitotoxic lesion of FN but not DN neurons abolishes stimulation-induced brain resistance to excitotoxic or ischemic damage [8].

The neuroprotective properties of subcortical stimulation are not unique for FN. Comparably, stimulation of the “subthalamic vasodilator area” (SVA) [12] or dorsal periaqueductal grey matter [13] similarly renders the brain tolerant to the ischemic damage. However, the time course and properties of the post-stimulation neuroprotection differ. Neuroprotective effect of SVA stimulation also lasts for over 10 days. However, it is most pronounced immediately after the stimulation slowly dissipating afterwards [12]. These observations allow to suggest the existence of systemic endogenous neuroprotective mechanisms, which are activated in response to adverse conditions [3,10].

Short period of global ischemia (2 min) significantly attenuates neuronal loss in hippocampus, striatum, cortex and thalamus induced by 10-min ischemia applied 2 days later in gerbils [14]. Comparable protective effect against cell loss following global 10 min ischemia after pre-exposure for 2–3 min of global ischemia was demonstrated in rats [15,16]. Infarction volume induced by transient (120 min) MCAO is smaller by ~60% if applied after exposure to 5 min global ischemia in rats without affecting cerebral perfusion [17]. The increased tolerance to damaging global ischemia following pre-exposure to short non-damaging global ischemia in vivo is a particular case of the more general phenomenon of preconditioning [18,19].

Preconditioning is the phenomenon of increased brain resistance to ischemia/hypoxia following pre-exposure of the animal to various factors [18]. Ischemic/hypoxic preconditioning or tolerance can be reproduced in different tissues and cell cultures and was suggested to be a universal phenomenon, which occurs in invertebrates and vertebrates [19]. Most of the research exploring the ischemic/hypoxic preconditioning is focused on the cellular mechanisms of preconditioning, e.g., see reviews [18,19,20]. While, undoubtedly, the end-effector mechanisms of preconditioning are cellular, the “initiators” [18] of neuroprotection may also involve systemic mechanisms. This suggestion finds support in the well-established phenomenon of “remote preconditioning”, when preconditioning stimulus, “initiator”, is applied at the sites remote from the protected organ/tissue [18,21]. We hypothesized that in vivo systemic endogenous neuroprotective mechanisms participate in the ischemic preconditioning triggered by global ischemia. To address this issue, we explored whether salvaging effect of global ischemic preconditioning can be affected by the excitotoxic lesion of intrinsic FN neurons, excitation of which induces neuroprotection [8]. Here we demonstrate that ablation of intrinsic FN neurons aggravated the effects of global brain ischemia and reversed the preconditioning protective effects of global ischemic preconditioning. The data suggest that integrative endogenous neuroprotective mechanisms are involved in intrinsic neuroprotection and development of the ischemia-induced tolerance in vivo.

## 2. Materials and Methods

All experiments were conducted in accord with the U.S. National Institutes of Health “Guide for the care and use of laboratory animals” and approved by the Institutional Animal Care and Use Committee of Mississippi University Medical Center, Jackson, MS. Experiments were performed in adult male Sprague-Dawley rats (250–300 g) (Charles River, Worcester, MA, USA). Animals were housed in the institutional animal facilities on 12 h day/night cycle with ad libitum access to food and water.

### 2.1. General Procedures

Procedures for surgery and microinjections have been detailed elsewhere [4,5,8] and are described here in brief. All procedures were performed in antiseptic conditions. The animals were anesthetized with isoflurane (induction 2.5% isoflurane in 80%/20% N_2_/O_2_ mixture in the induction chamber and 1.5–1.75% isoflurane was used for the maintenance), intubated and artificially ventilated. After anesthetization and intubation, the femoral artery was cannulated to record arterial blood pressure and to sample blood for pH, PaO_2_, PaCO_2_, and glucose (see Table 1). Body temperature was maintained at 37 °C using thermo blanket and rectal probe feedback.

### 2.2. Four-Vessel Occlusion

Four-vessel occlusion (4-VO) model of global ischemia was used [9,22]. Three days before the experiment, the animals were placed in a stereotaxic apparatus, after exposure of the first two cervical vertebrae the right and left *foramina alarae* of the first cervical vertebra were exposed and both vertebral arteries were cauterized with electrocautery needle 0.5 mm in diameter. Animals were allowed to recover. On the day of experiment, both carotid arteries were exposed through the midline cut while care was taken to avoid nerve damage. Surgical vascular microclips were applied on both arteries for the respective time. After the surgery clips were removed, wounds were closed and animals were allowed to recover.

### 2.3. Regional Cortical CBF

Regional cortical CBF was measured using laser Doppler flowmeter. After placing the animal in the stereotaxic frame, the dorsal surface of the skull was exposed through the midline cut. The bone over the parietal cortex was thinned to the *lamina vitrea* by shaving of the skull bone with the dental burr while the skull was irrigated with the saline to prevent overheating of underlying cortex. To position the laser Doppler needle probe (0.45 mm diameter) over the cortex, the site without visible large vessels was chosen within the square (2 × 2 mm) of thinned bone. A drop of mineral oil was placed under the probe to provide optical contact.

### 2.4. Fastigial Neurons Lesion

Intrinsic neurons of FN were selectively destroyed by microinjection of IBO as described earlier [8] five days before the experiment. IBO or its vehicle (PBS 0.1 M, pH 7.3) was microinjected into the cerebellar nuclei through capillary glass pipettes (~50 μm tip outer diameter). After exposure of the dorsal surface of the calvarium and occipital bone, the holes were drilled in the interparietal bone. To target the FN, the injection pipette was inserted with reference to the *calamus scriptorius* as stereotaxic zero: anterior 5 mm, lateral 0.8 mm, and dorsal 1.6 mm [8]. IBO or vehicle was injected over 3 min by hand. After injections, the pipette was removed and reinserted. Three injections were made on each side to destroy FN neurons. Lesions were placed at sites 0.2 and 0.4 mm caudal to the initial site. The total dose of 23 nmol was injected at each side (3.89 nmol in 30 nL/injection). The group of animals with equal volumes of PBS microinjected into six sites in FN constituted sham-lesioned controls. After the experiments, histological evaluation of localization and completeness of the cerebellar FN was done using hematoxylin/eosin (HE) stained slices (Figure 1). The spread of lesion (gliotic scar) was outlined on the appropriate levels of the stereotaxic atlas [23] (Figure 1).

### 2.5. Histological Processing

Seven days later after global ischemia, the animals were deeply anesthetized and perfused intracardially with 0.1M PBS followed by 4% paraformaldehyde in PBS. The brains were removed, postfixed in neutral-buffered paraformaldehyde for 2 days and embedded in paraffin and sliced at 8 μm. After deparaffinization, alternate sections were stained HE and thionin. We focused on changes of neurons of the Cornu Ammoni area 1 (CA1) region of the dorsal hippocampus from −3.3 to −4.3 mm from bregma. Hippocampal CA1 area neurons were examined on HE-stained slices at ϗ40 magnification. Normal neurons (nuclei) were counted and averaged across 3 fields on each side at 3 levels (300 μm apart) and average number of neurons/100 μm of the pyramidal cell layer was calculated. Neurons with swollen or shrunken perikarya, shrunken and darkly stained nuclei, and an acidophilic cytoplasm stained pink with eosin were defined as expressing ischemic damage and not counted. The “counter” researcher was blinded to the nature of the sample. Average number of CA1 pyramidal neurons/100 μm was calculated for each animal.

### 2.6. Experiments and Experimental Groups

All experimental animals (*n* = 64) underwent the comparable basic surgical preparation: bilateral coagulation of the vertebral arteries, exposure of common carotid arteries, intraparenchymal (FN or DN) microinjections (PBS or IBO) or just insertion and withdrawal of micropipette into FN. Similarly, prepared animals were randomly assigned to one of the following groups. Group 1: “Naïve”, just insertion of micropipette into FN 5 days before, animals received “sham” 20 min 4-VO followed by 7 days survival (*n* = 5), “Control”. Group 2: “Naïve” with injection of 90 nL of PBS in FN 5 days before animals received 20 min 4-VO followed by 7 days survival (*n* = 8), “4-VO”. Group 3: animals received IBO microinjection into FN 5 days before 20 min 4-VO followed by 7 days survival (*n* = 19),”IBO=>FN+4VO”. Group 4: “Naïve” animals underwent 5 min 4-VO followed 3 days later by 20 min 4-VO, followed by 7 days survival (*n* = 5), “PrCon+4VO”. Group 5: animals received IBO microinjection into FN followed 5 days later by 5 min 4-VO followed 3 days later by 20 min 4-VO followed by 7 days survival (*n* = 20), IBO=>FN+PrCon+4VO”. Group 6: animals received IBO microinjection into DN followed 5 days later by 5 min 4-VO followed 3 days later by 20 min 4-VO followed by 7 days survival (*n* = 7), “IBO=>DN+PrCon+4VO”.

### 2.7. Statistical Procedures

Data presented as mean ± standard deviation (SD). After testing for normality of distribution, one-way ANOVA analysis and Sheffe test were used to determine significance of differences between the groups. Chi-square test was used to compare differences in mortality.

## 3. Results

In the first two groups of animals we explored the effects of 4-VO global ischemia compared to “sham ischemia” on the integrity of neurons in the hippocampal area CA1. In the first, “sham”-occluded, group of animals (*n* = 5) no changes in CBF were observed. One week after the sham occlusion, the average number of cells per 100 μm was 19 ± 0.78 (averaged 90 fields of view) (Figure 2A and Figure 3). Acute occlusion of common carotid arteries in the second group triggered drop in CBF by 72 ± 1.2% and flattening of electroencephalogram (EEG) [9] as we described in our previous publication [9]. After 20 min of global ischemia, animals were allowed to recover and seven days later underwent histological analysis. Two out of eight animals (25%) did not survive for seven days and were excluded from the analysis. Following 20 min of ischemia, the number of CA1 cells significantly (*p* < 0.001) decreased by 34% compared to sham animals to 13 ± 1.3 cells/100 μm (averaged 108 fields of view) (Figure 2B and Figure 3).

In the third group of animals we explored whether the excitotoxic lesion of FN 5 days before 4-VO affects the severity of the damage of hippocampal cells induced by global ischemia. Twenty minutes of global ischemia in animals, in which FN was ablated, led to reduction of the number of CA1 neurons by 56% compared to sham-occluded animals to 8 ± 2.7 cells/100 μm (*n* = 5, 90 fields of view) (Figure 2C and Figure 3). Importantly, loss of cells following 20-min global ischemia was significantly higher than the cell loss in animals in which FN was not lesioned (*p* = 0.006). Moreover, lethality in FN-lesioned animals following 4-VO increased and reached 74% (5 out of 19 survived 7 days), which was significantly (χ^2^(1) = 5.53, *p* = 0.019) higher than in non-lesioned animals (25%) (Figure 4). These data strongly suggest that selective lesioning of FN neurons renders brain and animals more susceptible to ischemic insult.

In the next group of animals, we explored whether 5 min global ischemia can precondition, i.e., increase ischemic tolerance, against damage induced by subsequent 20 min global ischemia. Five-minutes preconditioning prevented neuronal loss in CA1 area triggered by 20 min global ischemia (*n* = 5, 20 ± 0.8 cells/100 μm, 90 fields of view) (Figure 2D and Figure 3). All five preconditioned animals survived 7 days following the 20-min global ischemia. Ablation of FN neurons preceding preconditioning reversed the salvaging effect of 5-min preconditioning. The cell loss following 20 min global ischemia in FN-lesioned preconditioned animals reached 45% (11 ± 2 cells/100 μm, *n* = 5, 90 fields of view, compared to preconditioned animals 20 ± 0.8 cells/100 μm, *p* < 0.001) (Figure 2E and Figure 3). FN lesion in preconditioned animals also significantly (χ^2^(1) = 9.38, *p* = 0.002) decreased 7 days post-occlusion survivability to 20% (Figure 4). In order to explore possible non-site-specific effect of intraparenchymal microinjections of IBO, comparable microinjection was made into DN. Excitotoxic lesion of DN non-significantly affected neuroprotective effect of ischemic preconditioning (*n* = 5, 17 ± 1.6 cells/100 um) (Figure 2F and Figure 3). Survival (5 out of 7) of DN-lesioned animals also was significantly higher than FN-lesioned animals (χ^2^(1) = 4.79, *p* = 0.029) (Figure 4). These findings strongly suggest the importance of the integrity of FN neurons for the expression of the ischemic preconditioning phenomenon and their role in the survivability of the global ischemia.

## 4. Discussion

Our experiments demonstrated that lesions of intrinsic FN neurons aggravate global ischemia-induced damage of CA1 hippocampal neurons, increasing it by 35%. Mortality rate of animals in which FN neurons were ablated increased by 41% compared to control non-lesioned group. These observations suggest that intrinsic FN neurons may play an important role in response to global ischemia and induction of tolerance against global cerebral ischemia. Short non-damaging global ischemia failed to induce tolerance to the following damaging global ischemia in animals, in which FN neurons were destroyed by local microinjection of ibotenic acid. This observation suggests that intrinsic FN neurons may participate in establishing tolerance triggered by global ischemic preconditioning.

Local microinjection of IBO induces acute excitotoxic neuronal damage [24] while relatively sparing fibers in passage. Histological examination of the injection sites in our experiments confirmed the destruction of local FN neurons. This allows to conclude that the effects of IBO microinjections into cerebellar FN were due to specific ablation of local neurons rather than non-specific effect of ibotenic acid administration, as destruction of DN neurons had no effects in our experiments.

The possible differences in degree of global ischemia also can be excluded as all animals demonstrated comparable decrease in CBF following four-vessel occlusion. Similarly, while isoflurane used in our experiments for anesthesia is known to exert neuroprotective and preconditioning effects [25,26], all animal groups were comparably anesthetized and the neuroprotective effect of isoflurane would equally affect all groups. Similarly, animals were artificially ventilated and blood gases were maintained at the comparable level and hence cannot explain the observed differences in responses to global ischemia in different experimental groups.

These considerations strongly suggest that observed differences in neuronal loss following global ischemia, survival of the animals, and modification of global ischemic preconditioning effects were due to specific lesion of intrinsic FN neurons.

Naturally occurring evolutionary ancient mechanisms for survival of low-oxygen/anoxic conditions [27,28,29] allow animals to survive very low levels of oxygen [30]. Besides long-term adjustments of body and brain metabolism related to long-term anoxia observed, e.g., in turtles or hibernating animals, the existence of the complex systemic mechanisms for an emergency response is also well documented. One of the examples is diving response, presented in all mammals [31,32,33]. Most individuums during lifetime face natural situations with decreased oxygen supply such as diving. Adjustments necessary to survive low-oxygen conditions well developed in diving animals exist in all mammals, including humans [34,35]. The diving response is a stereotypical systemic physiological change [31,32,35,36] geared towards the survival of hypoxic/adverse conditions known as “oxygen conserving reflex”. The function of these mechanisms is to adjust brain activity and metabolism to maintain brain survival before pathological events due to hypoxia/anoxia. These emergency protecting responses can be triggered either by external stimuli, like trigeminal nerve stimulation [37], or by excitation of the brain chemo- and oxygen sensing neurons [38,39,40,41], which respond to the decreasing levels of oxygen before it drops to the levels affecting cells metabolism [41]. This early warning system initiates systemic responses in advance of injurious developments “preconditioning” the brain, i.e., rendering it tolerant to ischemia/hypoxia. While most of the research related, for example, to the mechanism of diving response or hibernation concentrated on the brain mechanisms of autonomic adjustments [31,32,33,37] or cellular mechanisms [42] forebrain/cerebellar integral mechanisms of emergency response to such events as acute hypoxia, ischemia remain largely unexplored.

Well-established phenomenon of ischemic preconditioning also can be considered as an adaptive response to dangerous situations, which trigger short- and long-term tolerance to damaging ischemia/hypoxia. The phenomenon of preconditioning, defined as increased tolerance to ischemia/hypoxia after pre-exposure to various stimuli received a lot of attention [18,19,43,44]. However, the majority of the studies of preconditioning address cellular mechanisms of adaptation to ischemia or direct effect of preconditioning stimuli on the cells, while systemic integrative mechanisms remain understudied.

In 1997, Reis and co-authors [10] theorized the existence of the endogenous neuroprotective system, which provides capabilities of the brain survival in the emergency adverse situation through various mechanisms. These mechanisms purportedly initiate multifaceted events, which increase brain tolerance to adverse conditions at systems and cellular levels. Based on our findings reported here, we postulate that endogenous neuroprotective system participates in systemic integrative mechanisms protective against acute ischemia/hypoxia, and is involved in establishing ischemic tolerance triggered by global ischemic preconditioning.

In the course of our studies of intrinsic neurogenic neuroprotection, the group of brain structures was identified, which when activated renders brain tolerant to global or focal ischemic damage [3,10]. One of the most explored structures so far is the cerebellar fastigial nucleus [1,2,3].

Cerebellar fastigial (medial) nucleus located near the midline at the roof of the fourth ventricle is phylogenetically oldest and evolutionary conserved cerebellar nucleus [45]. Neurons of cerebellar FN are heterogeneous and include glutamatergic, GABAergic and glycinergic neurons [46,47,48].

Besides cerebellar cortical and olivar projections, FN neurons receive multiple afferent projections from the posterior hypothalamus, pontine nuclei and the nucleus *reticularis tegmenti pontis*, the lateral reticular nucleus, the raphe nuclei, and the spinal cord [49,50,51,52,53,54].

In turn, FN neurons issue multiple efferent projections to various structures including medial reticular formation, nucleus reticularis gigantocellularis, magnocellularis, paramedian reticular nucleus, parasolitary region, basilar pontine nucleus, mesencephalon, visual thalamus, ventromedial and ventrolateral thalamus, dorsomedial and paraventricular hypothalamic nuclei as well as to forebrain, including limbic system [55,56,57,58,59,60,61,62,63,64]. Due to the wide connections, FN plays an important role not only in vestibulo- and oculomotor functions, but also in various non-motor functions including endocrine, immune, cardiovascular, respiratory, behavioral and cognitive functions [10,65,66,67,68,69,70].

In our experiments, chemical ablation of FN neurons decreased animals’ survival following global ischemia and aggravated hippocampal neuronal damage. These observations are in line with the previously demonstrated negative effect on survivability of dogs following the hemorrhage after the electrolytic lesion of FN [71]. Because of comparable negative effect on survivability of rats and dogs after chemical or electrolytic lesions of FN, it is possible to speculate that activity of FN neurons may be critical for survival of damaging events. Severe ischemia induced by four-vessel occlusion or hemorrhage may excite FN neurons through activation of oxygen sensing neurons located in the posterior hypothalamus [72,73] or rostral ventrolateral medulla [49]. The proper neurons of FN are also considered to be chemosensitive [39,74]. Thus, it is conceivable that FN neurons could be activated in response to hypoxia/ischemia and initiate systemic protective response, which would include autonomic, immune and cellular adjustments targeted to survival.

Attenuation of global ischemia-induced neuroprotection following the lesion of FN neurons in our experiments allows to speculate that their activity is necessary for the triggering of protective effect. Unfortunately, most of the experiments, which demonstrated neuroprotective properties of FN stimulation, were performed using electrical stimulation, which non-selectively excites proper FN neurons and fibers in passage. However, some data allow to distinguish effects of electrical and chemical stimulation of FN, which could produce opposite effects. Electrical stimulation, which also excites fibers of passage (see [64,75]) along with neuronal excitation of FN produces well-described fastigial pressor response [76,77,78] accompanied by global metabolically independent increase in CBF. However, chemical stimulation with glutamate or acetylcholine produces an opposite effect, known as fastigial depressor response [45,79], which most probably mediated by the excitation of the proper FN neurons. Chemical stimulation of FN also induces global brain metabolic suppression (reduction of glucose utilization) accompanied by decrease in CBF, possibly due to decreased metabolism [80]. Electrical stimulation of FN decreases neuronal excitability [10], presumably through the opening of potassium channels [8,81,82]. FN stimulation slows EEG activity [83,84] and suppresses seizure activity [85]. These observations suggest decrease in brain activity, which, in turn, increases its tolerance to acute ischemia [30], promoting survival of the brain cells and organism.

It is well established that short global ischemia induces long-term ischemic tolerance [18,19,43]. Our present experiments demonstrated that excitotoxic lesion of FN significantly attenuates induced by short global ischemia tolerance to delayed neuronal death. Electrical stimulation of FN is known to produce long-lasting neurogenic preconditioning [1,2,3] including attenuation of delayed neuronal death [9]. While the specific mechanisms of FN stimulation-induced neuroprotection remain unknown, there are some protective mechanisms that seem to be comparable to the mechanisms of global ischemic preconditioning.

Opening of potassium channels following FN stimulation [81,86] is a necessary condition for the expression of long-term ischemic tolerance [3]. This observation is comparable with the observation that ischemic tolerance following global ischemic preconditioning also depends on the opening of potassium channels [16]. Delayed hippocampal neuronal death following ischemic insult occurs preferentially through mitochondrial apoptotic pathway [87,88,89]. Stimulation of FN has been shown to stabilize mitochondria and to increase mitochondrial tolerance to apoptotic stimuli [90,91,92]. The increased mitochondrial tolerance after FN-induced preconditioning is comparable to the attenuation of mitochondrial apoptosis following global ischemic preconditioning [93,94,95].

The maximum effect of stimulation is observed 72 h after the stimulation [96] suggesting that neuroprotective effect of FN stimulation requires changes in gene expression. Indeed, FN stimulation changes in miRNA expression [97,98] and upregulation of PPAR-γ [99,100], which, in turn exerts antiapoptotic effect [101]. Global ischemic preconditioning modifies gene expression, presumably promoting ischemia tolerant phenotype, including changes in expression of genes involved in immune response [102]. Immune mechanisms play an important role in establishing ischemic tolerance at the systems level [44]. FN also seems to be involved in the stressors-induced modulation of immune responses. Stimulation of FN not only increases resistance of cerebral microvessels to inflammatory mediators [103,104,105] but also exerts systemic immunomodulatory effects through cerebello-hypothalamic projections [106,107]. These observations suggest that FN-induced preconditioning and global ischemia-induced preconditioning may exert their neuroprotective effect through various common end-effector mechanisms.

The effects of FN excitation also have other protective effects including alleviation of post-stroke depression [108,109] and promotion of survival of transplanted stem cells [110,111]. Moreover, stimulation of FN cells may even protect against myocardial infarction [112].

Our data along with these observations allow to suggest that excitation of FN neurons can trigger complex concerted functional adjustments, which integrate autonomic, endocrine, immune and cellular short- and long-term modifications targeted towards survival of adverse conditions, ischemia/hypoxia, in particular. Based on the shared mechanisms between FN-induced and global ischemia-induced preconditioning and our observations that ablation of FN neurons attenuates neuroprotective effects of global ischemic preconditioning, it is possible to hypothesize that at least some of the neuroprotective effects triggered by global ischemic preconditioning may depend on the activation of FN neurons.

## 5. Conclusions

Our current findings further support the hypothesis of the existence of the intrinsic endogenous neuroprotective system and the FN as a part of it. Numerous attempts to identify the “golden bullet” to treat stroke by identifying the single target, unfortunately, were not satisfactory. Powerful endogenous mechanisms have developed in evolution to protect the brain against adverse conditions such as lack of oxygen. These mechanisms are a sum of highly coordinated physiological adjustments occurring at all levels: from the level of whole organism to the cellular level. We suggest that FN is a part of this neuroprotective system. Further exploration and understanding of the functioning of endogenous neuroprotective system will allow to identify methods of amplifying the efficiency of endogenous neuroprotective mechanisms and thus to develop new therapeutic approaches for the treatment of stroke, which will be able to amplify the integrative neuroprotective response. Changes of strategic approach from targeting single pathway to activation of multifaceted multilevel integrative mechanisms may allow developing new therapeutics based on the different principle of employing endogenous neuroprotective system instead.

## Figures and Tables

**Figure 1 brainsci-07-00121-f001:**
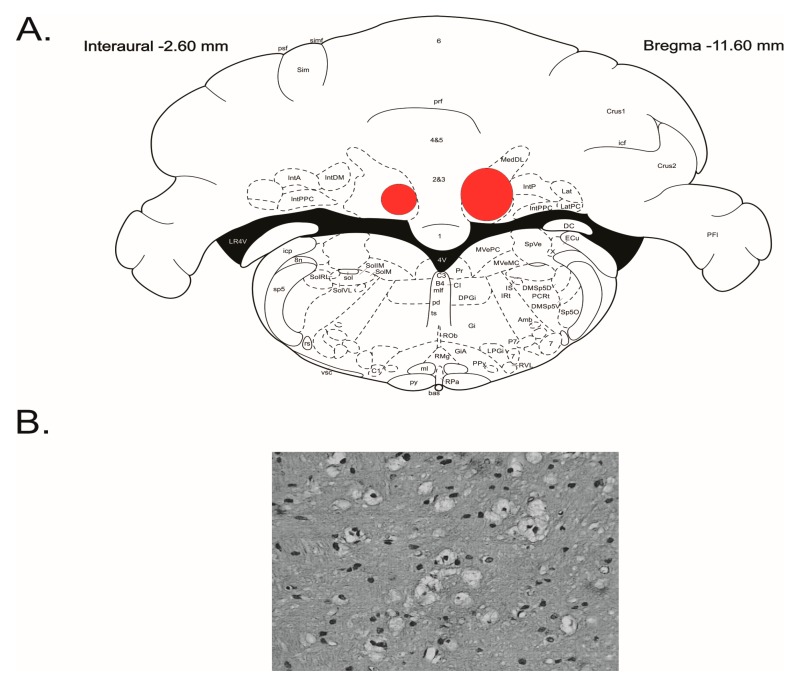
Summarized localization of the lesioned areas in the cerebellar fastigial nucleus. (**A**) Outer borders of overlapped outlines of histological damage 5 days following the injection of IBO depicted on the appropriate level of stereotaxic atlas [23]. (**B**) Gliosis at the IBO injection site.

**Figure 2 brainsci-07-00121-f002:**
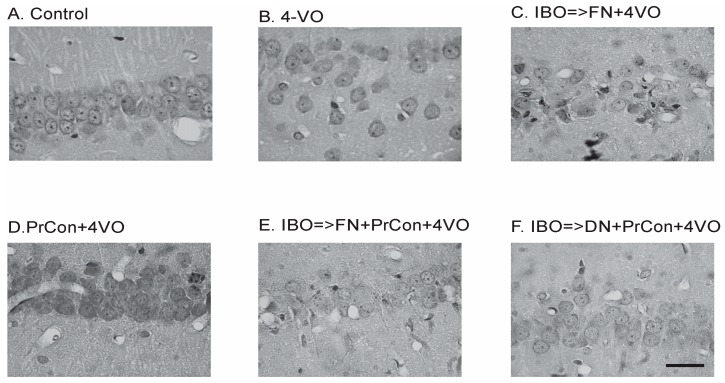
Changes of the hippocampal Cornu Ammoni area 1 (CA1) pyramidal layer neurons 7 days after sham or 4-vessels occlusion (4-VO) in different experiments. (**A**) CA1 neurons in animal, which underwent PBS injection into fastigial nucleus (FN) and sham 4-VO; (**B**) CA1 neurons following 20-min 4-VO; (**C**) CA1 neurons in the animal, which received microinjection of ibotenic acid (IBO) into fastigial nucleus five days before 4-VO; (**D**) CA1 neurons in the animal which underwent 5-min 4-VO three days before the 20-min 4-VO; (**E**) CA1 neurons in the animal, which received microinjection of IBO into FN 5 days before the 5-min 4-VO followed by 20-min 4-VO three days later; (**F**) CA1 neurons in the animal which received microinjection of IBO into DN 5 days before the 5-min 4-VO followed by 20-min VO three days later. Bar of panel F equals 50 μm. Other panels have the same magnification.

**Figure 3 brainsci-07-00121-f003:**
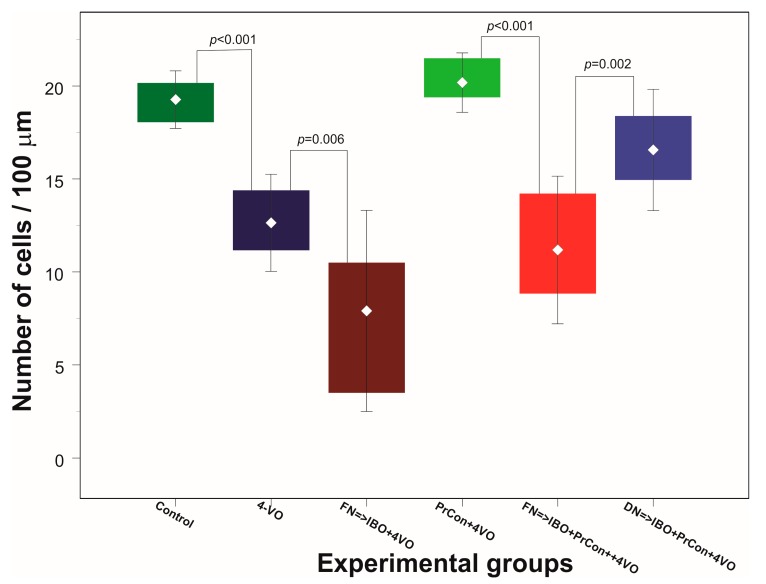
Changes of the average number of hippocampal CA1 pyramidal layer neurons 7 days after sham or 4-vessels occlusion (4-VO) in different experiments. Dark green box: CA1 neurons in animals, which underwent PBS injection into fastigial nucleus (FN) and sham 4-VO (*n* = 5); Dark blue boxes. CA1 neurons following 20-min 4-VO (6); Brown box: CA1 neurons in the animals, which received microinjection of ibotenic acid into (IBO) fastigial nucleus five days before 4-VO (*n* = 5); Light green box: CA1 neurons in the animals which underwent 5-min 4-VO three days before the 20-min 4-VO (*n* = 5); Red box: CA1 neurons in the animal, which received microinjection of IBO into FN 5 days before the 5-min 4-VO followed by 20-min 4-VO three days later (*n* = 5); Light blue box: CA1 neurons in the animal which received microinjection of IBO into DN 5 days before the 5-min 4-VO followed by 20-min VO three days later (*n* = 5). Boxes indicate minimum–maximum spread; middle rhombus indicates mean; whiskers indicate 2 × SD. *p* indicate significance of the difference (One-way ANOVA *F* (5, 25) = 42.632, *p* < 0.001, Sheffe test).

**Figure 4 brainsci-07-00121-f004:**
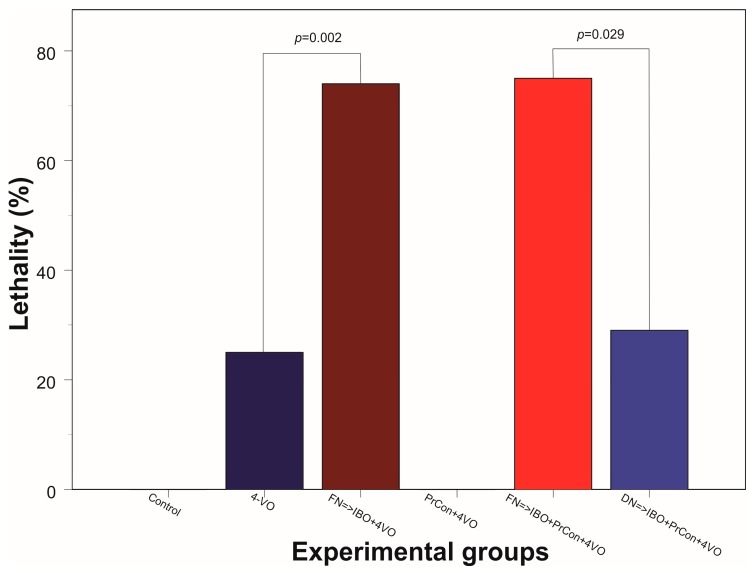
Animal lethality within 7 days following 20-min 4 vessel occlusion in different experiments. Control group did not have lethal cases (0/5); Dark blue bar: lethality within one week following 20-min 4-VO (2/8); Brown bar: lethality in the animals, which received microinjection of ibotenic acid into (IBO) fastigial nucleus five days before 4-VO (14/19); PrCon + 4-VO: lethality in the animals which underwent 5-min 4-VO three days before the 20-min 4-VO (0/5); Red bar: lethality in the animals, which received microinjection of IBO into FN 5 days before the 5-min 4-VO followed by 20-min 4-VO three days later (15/20); Light blue box: lethality in the animals which received microinjection of IBO into DN 5 days before the 5-min 4-VO followed by 20-min VO three days later (2/7). *p*—significance χ^2^.

**Table 1 brainsci-07-00121-t001:** Blood gases and glucose content in the blood of animals in all experimental groups immediately after the initiation of the 20 min 4-vessel global occlusion. Groups are designated as in text: Group 1: “Naïve”, just insertion of micropipette into fastigial nucleus (FN) 5 days before, and “sham” 20 min four-vessel occlusion (4-VO). Group 2: “Naïve” with injection of 90 nL of PBS in FN 5 days before 20 min 4-VO. Group 3: animals received ibotenic acid (IBO) microinjection into FN 5 days before 20 min 4-VO. Group 4: “Naïve” animals underwent 5 min 4-VO followed 3 days later by 20 min 4-VO. Group 5: animals received IBO microinjection into FN followed 5 days later by 5 min 4-VO followed 3 days later by 20 min 4-VO. Group 6: animals received IBO microinjection into dentate (lateral) nucleus (DN) followed 5 days later by 5 min 4-VO, followed 3 days later by 20 min 4-VO. Data present as average ± standard deviation (SD). One-way ANOVA (Analysis of Variance) was used for the comparison of parameters between all experimental groups. No groups were significantly different. CBF: cerebral blood flow.

Group	paCO_2_ (mmHg)	paO_2_ (mmHg)	pH	Glucose (mg/dL)	CBF Drop (% from Baseline)
1. Naïve, sham 4-VO	41.0 ± 6.4	111.8 ± 15.1	7.4 ± 0.1	117.1 ± 16.2	0
2. 20 min 4-VO	39.8 ± 9.0	122.6 ± 15.4	7.4 ± 0.1	116.1 ± 19.3	72.1 ± 1.5
3. IBO + 20 min 4-VO	38.7 ± 10.6	112.3 ± 11.0	7.4 ± 0.1	127.2 ± 16.5	72.3 ± 1.7
4. Precondition + 4-VO	47.6 ± 8.0	102.6 ± 16.0	7.3 ± 0.1	127.4 ± 22.5	72.5 ± 1.5
5. IBO ≥ FN+PrCon + 4-VO	44.6 ± 12.6	113.8 ± 17.6	7.3 ± 0.1	117.7 ± 15.1	72.5 ± 1.2
6. IBO ≥ DN+PrCon + 4-VO	44.3 ± 10.7	110.0 ± 28.4	7.3 ± 0.1	104.6 ± 12.5	73.6 ± 1.2
ANOVA F(5.59)	1.004, *p* = 0.424	0.969, *p* = 0.444	1.917, *p* = 0.105	1.831, *p* = 0.121	1.067, *p* = 0.388 excluding sham

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
