# Peer review of "Integrity of Cerebellar Fastigial Nucleus Intrinsic Neurons Is Critical for the Global Ischemic Preconditioning"

_brainsci, 2017, doi:10.3390/brainsci7100121_

Round 1

Reviewer 1 Report

Dear Authors, It is a nicely summed up manuscript. Congratulations! I am sure the manuscript would be of interest to several researchers of relevance especially stroke related research. The authors suggest that FN could be play one of the key roles in the proposed neuroprotective system. However, there are few suggestions and comments to be considered that I highlighted before the manuscript is “accepted” for a publication. Please find below the suggestions, at the end of this message. All the best! Thanks Comments and Suggestions to the Authors Abstract: 1) line 19: change to “…….before exposure to 20 min 4 vessel occlusion…..” OR rephrase the entire sentence. Introduction: 2) line 77: change to “….develpment of the ischemia-induced tolerance in vivo.” Results: 3) line 163: abbreviate “CBF”, when it is used first 4) give space before and after “±” throughout the manuscript 5) change to “(fig 2B and 3 or fig 2B & fig 3)” throughout the manuscript, wherever two figures are mentioned. 6) Figure 2 legend: please mention the scale bar at least in one of the images. 7) Figures 3 and 4: please rename the “X-axis” with a more appropriate word than stating “experiment 8) line 218: replace “there” with “their” 9) line 220: change to “Figure 4” Discussion: 10) line 230: change to “aggravate” 11) line 231-232: change to “Mortality rate of animals in which FN neurons was ablated increased by 41% compared to control non-lesioned group.” 12) line 234-236: rephrase the sentence 13) line 263: change to “towards” 14) line 264: delete “an” 15) line 265: delete “take place” 16) line 276: change to “situations” and “trigger” 17) line 334: change to “present” and “attenuate” 18) line 338: change to “….remains unknown, there are some…..mechanisms that seem to be comparable……” 19) line 350: change to “miRNA expression” 20) line 353: change to “play” 21) line 365: change to “towards” 22) line 368: change to “hypothesize” Thanks

Author Response

Dear Reviewer,

Thank you very much for your interest in our work and your kind words.

In the revised version of the manuscript we implemented all your suggestions ,, which undoubtedly improve the manuscript.

We sincerely appreciate your time and thoroughness.

The following changes have been made:

1) line 19: change to “…….before exposure to 20 min 4 vessel occlusion…..” OR rephrase the entire sentence. Introduction: Corrected

2) line 77: change to “….development of the ischemia-induced tolerance in vivo.”  changed

Results:

3) line 163: abbreviate “CBF”, when it is used first CBF abbreviation is defined on line 40

4) give space before and after “±” throughout the manuscript  Has been done

5) change to “(fig 2B and 3 or fig 2B & fig 3)” throughout the manuscript, wherever two figures are mentioned. Done

6) Figure 2 legend: please mention the scale bar at least in one of the images. Bar has been added

7) Figures 3 and 4: please rename the “X-axis” with a more appropriate word than stating “experiment Done

8) line 218: replace “there” with “their” Done

9) line 220: change to “Figure 4” Done

Discussion:

10) line 230: change to “aggravate” Done

11) line 231-232: change to “Mortality rate of animals in which FN neurons was ablated increased by 41% compared to control non-lesioned group.” DONE

12) line 234-236: rephrase the sentence Done

13) line 263: change to “towards” Done

14) line 264: delete “an” Done

15) line 265: delete “take place” Done

16) line 276: change to “situations” and “trigger” Done

17) line 334: change to “present” and “attenuate” Done

18) line 338: change to “….remains unknown, there are some…..mechanisms that seem to be comparable……” Done

19) line 350: change to “miRNA expression” Done

20) line 353: change to “play” Done

21) line 365: change to “towards” Done

22) line 368: change to “hypothesize” Done

Reviewer 2 Report

In the article titled “Integrity of cerebellar fastigial nucleus intrinsic neurons is critical for the global ischemic preconditioning” the author’s investigated the role of the fastigial nucleus (FN) in ischemic preconditioning. The author’s data suggest that ablation of the FN results in a decrease in survival following four vessel occlusion and the loss of ischemic preconditioning. Interestingly, this increase in damage was not present in animals where the cerebellar dentate nucleus was damaged. Overall, the data suggest that the FN is one of area of the brain required for ischemic preconditioning.

This review has some minor comments:

1.    The weight of the animals is currently listed as 250+-350g. I am assuming that this is incorrect.

2.    Is the damage following consistent across animals? That is, was the injected/damaged area calculated to determine that the damage following injection was similar for each animal injected?

3.    Scale bars are not present on the images in Fig. 2.

4.    Is the labeling in Figure 3 for DNàIBO+4VO incorrect? Shouldn’t this also indicate preconditioning?

5.    The authors indicate that decrease in blood flow and blood gases were similar in all groups; however, it would be helpful to insert a table with these values and utilizing statistical analysis to confirm that there is no difference between the groups. 

Author Response

Dear Reviewer,

Thank you very much for your time, comments and suggestions, which undoubtedly will improve the manuscript.

We implemented all your suggestions.

1.    The weight of the animals is currently listed as 250+-350g. I am assuming that this is incorrect. Sorry it was a typo, and it was corrected.

2.    Is the damage following consistent across animals? That is, was the injected/damaged area calculated to determine that the damage following injection was similar for each animal injected?

Unfortunately it is difficult to outline the damage exactly. We contoured the obvious area of gliosis, but it is not absolutely precise. We overlapped all contours and the outer outline of ALL gliotic areas in all animals are reflected on the diagram. The lesion mostly located in the ventro-rostral area of the fastigial nucleus. It is also important that we injected the same volume of ibotenic acid in all animals, which make us confident that the  most active area of FN, which was shown in the previous work to induce protective effect, was lesioned. However, we cannot exclude that some "spillover" occur into the nearby parenchyma, but most probably it was not systematic and did not affect the main conclusions of our study. We completely agree that it would be very interesting and important to try to correlate the exact localization of the lesion with the outcome. In the future we plan to do stereometric work with the specific markers to identify the exact localization of the crucial FN area and especially neurons responsible for neuroprotection.   

3.    Scale bars are not present on the images in Fig. 2. Thank you for bringing it to our attention. We placed the scale bar on the figure.

4.    Is the labeling in Figure 3 for DNàIBO+4VO incorrect? Shouldn’t this also indicate preconditioning? Thank you to bring it to our attention and we apologize for the oversight. Appropriate designation has been made.

5.    The authors indicate that decrease in blood flow and blood gases were similar in all groups; however, it would be helpful to insert a table with these values and utilizing statistical analysis to confirm that there is no difference between the groups.

It is an important information and we included the table (Table 1) with the basic parameters in each group at the moment when 4VO was applied. One-way ANOVA analysis demonstrated that the groups did not differ in the initial parameters.